# The Generation of Attenuated Mutants of East Asian Passiflora Virus via Deletion and Mutation in the N-Terminal Region of the HC-Pro Gene for Control through Cross-Protection

**DOI:** 10.3390/v16081231

**Published:** 2024-07-31

**Authors:** Duy-Hung Do, Xuan-Tung Ngo, Shyi-Dong Yeh

**Affiliations:** 1Department of Plant Pathology, National Chung Hsing University, Taichung 402, Taiwan; duyhung6489@gmail.com; 2Plant Pathology Division, Plant Protection Research Institute, Hanoi 10000, Vietnam; 3Department of Horticulture, National Chung Hsing University, Taichung 402, Taiwan; tungngo.vrq@gmail.com; 4Advanced Plant and Food Crops Biotechnology Center, National Chung Hsing University, Taichung 402, Taiwan; 5Overseas Vietnam Agricultural Science and Technology Innovation Center, National Chung Hsing University, Taichung 402, Taiwan

**Keywords:** potyvirus, East Asian Passiflora virus, HC-Pro, cross-protection, attenuated mutant

## Abstract

East Asian Passiflora virus (EAPV) causes passionfruit woodiness disease, a major threat limiting passionfruit production in eastern Asia, including Taiwan and Vietnam. In this study, an infectious cDNA clone of a Taiwanese severe isolate EAPV-TW was tagged with a green fluorescent protein (GFP) reporter to monitor the virus in plants. *Nicotiana benthamiana* and yellow passionfruit plants inoculated with the construct showed typical symptoms of EAPV-TW. Based on our previous studies on pathogenicity determinants of potyviral HC-Pros, a deletion of six amino acids (d6) alone and its association with a point mutation (F_8_I, simplified as I_8_) were conducted in the N-terminal region of the HC-Pro gene of EAPV-TW to generate mutants of EAPV-d6 and EAPV-d6I_8_, respectively. The mutant EAPV-d6I_8_ caused infection without conspicuous symptoms in *N. benthamiana* and yellow passionfruit plants, while EAPV-d6 still induced slight leaf mottling. EAPV-d6I_8_ was stable after six passages under greenhouse conditions and displayed a zigzag pattern of virus accumulation, typical of a beneficial protective virus. The cross-protection effectiveness of EAPV-d6I_8_ was evaluated in both *N. benthamiana* and yellow passionfruit plants under greenhouse conditions. EAPV-d6I_8_ conferred complete cross-protection (100%) against the wild-type EAPV-TW-GFP in both *N. benthamiana* and yellow passionfruit plants, as verified by no severe symptoms, no fluorescent signals, and PCR-negative status for GFP. Furthermore, EAPV-d6I_8_ also provided complete protection against Vietnam’s severe strain EAPV-GL1 in yellow passionfruit plants. Our results indicate that the attenuated mutant EAPV-d6I_8_ has great potential to control EAPV in Taiwan and Vietnam via cross-protection.

## 1. Introduction

Passionfruit (*Passiflora edulis* Sims), a native plant of Brazil, is an economically important fruit crop in tropical and subtropical regions [1]. However, passionfruit production is seriously limited worldwide by passionfruit woodiness disease (PWD), including in Taiwan and Vietnam [2,3]. Passionfruit plants affected by PWD show a wide range of symptoms, from leaf mosaic to woodiness and the deformation of fruits. The causal agent of PWD was identified as a few positive-sense RNA viruses belonging to the genus *Potyvirus* [4]. The first reported potyvirus infecting passionfruit was the passionfruit woodiness virus (PWV) in Australia [5,6]. A potyvirus distinct from PWV in South Africa and named South African Passiflora virus (SAPV) was described [7]. The virus was later identified as cowpea aphid-borne mosaic virus (CABMV) [8] and was also reported to infect passionfruit in Brazil [9]. In Uganda, PWD was caused by another potyvirus, the Ugandan Passiflora virus (UPV) [10]. In Asia, potyvirus-causing PWD has been identified as East Asian Passiflora virus (EAPV) in Japan [11], Taiwan [2], and China [12]. Another potyvirus-causing PWD is Telosma mosaic virus (TelMV), which has prevailed in Thailand and China [13,14]. A study in Japan has discovered a new potyvirus known as East Asian Passiflora distortion virus (EAPDV), which is also responsible for foliar mosaic and fruit malformation in *Passiflora* species [15]. A recent study found that EAPV, TelMV, and a new potyvirus named Passiflora mottle virus (PaMoV) are three causal agents of PWD in Vietnam [3].

EAPV causes PWD, characterized by leaf mosaic, fruit malformation and woodiness, and stunted growth [16]. In Japan, EAPV is associated with PWD, which was misidentified as a strain of PWV based on its symptomatology. In 2006, two complete genomic sequences of EAPV-AO and EAPV-IB from Japan were reported [11,17]. These EAPV isolates share 67–72% of the aa sequence identity of the CP with those of PWV isolates, indicating that EAPV is a distinct potyviral species from PWV [17]. In addition, the aa sequences of these two EAPV strains share 83% of their identities, indicating that the two viruses are different EAPV strains [17]. In Taiwan, the causal agent of PWD was also misidentified as PWV for decades [18,19] but has been reclassified as EAPV after comparing genomic sequences [2]. EAPV was also reported to infect passionfruit in Malaysia [20], China [12], and Vietnam [3].

Cross-protection is a phenomenon in which plants infected with a mild or attenuated strain of a virus provide protection against subsequent infection by related severe strains of the same virus [5,21,22]. Many viruses have been controlled by cross-protection for decades. Noteworthy examples of large-scale applications include tobacco mosaic virus [23,24], papaya ringspot virus [25], citrus tristeza virus [26], zucchini yellow mosaic virus (ZYMV) [27,28], and pepino mosaic virus [29]. Cross-protection has also been applied to passionfruits for controlling PWV in Australia [30] and Brazil [31].

The helper component protease (HC-Pro) is an essential and well-characterized multifunctional protein of potyviruses, involved in the viral movement, pathogenicity, viral transmission by aphids, polyprotein maturation, and RNA-silencing suppression [32,33]. Modifying the HC-Pro gene of potyviruses can reduce its RNA-silencing suppression (RSS) capability and generate useful mild strains for cross-protection [28,34,35]. The conserved motif FR_180_NK of HC-Pro plays a critical role in potyvirus virulence and RSS ability [28,34,36]. The mutated strain ZYMV-AG (R_180_I) from an Israel strain [36] and ZYMV-AC double mutant (R_180_I and E_396_N) from a Taiwan strain [28] induced infection without conspicuous symptoms and protected hosts against severe viruses. In turnip mosaic virus (TuMV) and ZYMV, the deletion of N-terminal six amino acids upstream the motif F_7_WKG coupled with F_7_I mutation induced attenuated symptoms, reduced RSS ability, and abolished aphid transmission ability [35]. The role of motif FWKG in α-helix 1 element was also studied to cause symptom attenuation in PRSV [37,38]. Additionally, in EAPV, F_8_ in FWKG motif (FFKG in EAPV), R_181_ in FRNK motif, and E_397_ residues were manipulated to generate two double-mutations EAPV-I_8_N_397_ and EAPV-I_181_N_397_, both infect *Nicotiana benthamiana* and passionfruit without apparent symptoms and provide protection against the wild-type EAPV [39].

In this investigation, to monitor the existence of the challenge virus in plants, we inserted the green fluorescent protein (GFP) reporter into a cDNA infectious clone of a severe Taiwan isolate EAPV-TW [39]. Site-directed mutagenesis was conducted to mutate the essential pathogenicity motifs F8FKG in the N-terminal region of EAPV. Six amino acids upstream of the FFKG motif were deleted (d6) or further coupled with the mutation F_8_→I (simplified as I_8_) to generate EAPV-d6I_8_ or EAPV-d6 mutants, respectively. The accumulation dynamics and the stability of the two mutants were analyzed in the plants of *N. benthamiana* to examine the characteristics of both mild mutants. Cross-protection effectiveness of EAPV-d6I_8_ was evaluated in *N. benthamiana* and yellow passionfruit plants, using GFP as a tag to monitor the severe strain after challenge inoculation. Our results showed that the attenuated mutant EAPV-d6I_8_ has great potential to control EAPV in Taiwan and Vietnam by cross-protection.

## 2. Materials and Methods

### 2.1. Virus Isolates and Culture

East Asian Passiflora virus isolates EAPV-TW from Taiwan [2] and EAPV-GL1 from Vietnam [3] were individually maintained in *N. benthamiana* plants. These isolates were transferred mechanically every four weeks and kept in a temperature-controlled greenhouse (25 to 28 °C) for further study.

### 2.2. Tagging of EAPV-TW with GFP

To monitor the infection of the severe strain EAPV-TW, the GFP and an NIa protease cleavage site (SVVVQ/S) were constructed in the infectious clone of EAPV-TW (pEAPV-TW, [39]) to generate pEAPV-TW-GFP. For this purpose, two restriction sites, *Sac*I in P1 and *Sna*BI in P3, were selected (Figure 1A). The P1 region downstream of *Sac*I was amplified by EAPV 2 fwd and P1HCPro rev primers, while the HC-Pro and P3 regions upstream of *Sna*BI were amplified by P1HCPro fwd and EAPV P3 rev primers (Table 1). The primers P1HCPro fwd and P1HCPro rev contained NIa protease cleavage site and two restriction sites *Sac*II and *Xma*I, respectively. Two fragments were fused by overlapping PCR using EAPV 2 fwd and EAPV P3 rev primers, and the product was then digested with *Sac*I and *Sna*BI. The obtained fragment was inserted into pEAPV-TW, which was also digested with SacI and SnaBI. The final clone was named pEAPV-MCS, harboring the multi-cloning sites and an NIa-Pro cleavage site (Figure 2A). The GFP-coding region was amplified from p35ZYMVGFPhis [40] using GFP primer pair GFP SacII fwd/GFP XmaI rev (Table 1), digested with *Sac*II and *Xma*I, and inserted into the polyprotein ORF between the P1 and HC-Pro cistrons (Figure 1A).

### 2.3. Infectivity Assay

Systemic host plants of *N. benthamiana* at the 5–6 cm stage were used to test the infectivity of cDNA clones. *E. coli* JM109 cells (Promega, Wisconsin) were transformed with the construct pEAPV-TW-GFP by heat shock according to the manufacturer’s instructions. The plasmid was then extracted using a plasmid miniprep purification kit (GeneMark, Taichung, Taiwan). An aliquot of 10 µL of plasmid (5 µg) was mechanically applied to each *N. benthamiana* leaf dusted with Carborundum (400 mesh). Later, the yellow passionfruit plants (*Passiflora edulis* f. *flavicarpa* Degener) with two fully expanded leaves were inoculated with the virus prepared from the infected *N. benthamiana* plants. The presence of the virus in plants inoculated with pEAPV-TW-GFP was checked via Western blotting using the described procedure [3] with EAPV CP antiserum [19] or GFP antiserum [40]. Extracts from equal amounts (0.01 g) of leaf tissue collected at ten days post-inoculation (dpi) were loaded, separated on an SDS-gel (12.5%), transferred onto a nitrocellulose membrane, and reacted with EAPV CP antiserum or GFP antiserum, both diluted at 1:10,000. The presence of GFP in inoculated plants was also verified by RT-PCR using primer pair EAPV 2 fwd and EAPV HCPro rev (Figure 2A, Table 1).

### 2.4. Generation of Attenuated EAPV Mutants by Modification of HC-Pro Gene

A fragment from P1 to P3 was amplified by RT-PCR using KOD-Plus-Neo polymerase (Toyobo, Osaka, Japan) with primer pair EAPV 2 fwd upstream of *Sac*I site and EAPV P3 rev downstream of *Sna*BI site (Table 1). The PCR product was eluted by the gel elution kit (Genmark, Taichung, Taiwan) and subcloned into pCR-Blunt II-TOPO vector (Invitrogen, CA). The intermediate clone was named pCR-EAPV-HCPro (Figure 3A). Site-directed mutagenesis was performed on pCR-EAPV-HCPro by circular amplification using KOD-Plus-Neo (Toyobo, Osaka, Japan), with the condition according to the manufacturer’s instruction. The primer pair EAPVd6 fwd and rev (Table 1) was used to delete six amino acids upstream of the F_8_FKG motif in the N-terminal region of HC-Pro to generate a mutant of EAPV-d6, while primer pair EAPVd6F8I fwd and rev (Table 1) deleted six amino acids coupled with the mutation F_8_I to generate a mutant of EAPV-d6I_8_. After mutagenesis, EAPV-d6 and EAPV-d6I_8_ were mechanically introduced into *N. benthamiana* and yellow passionfruit plants following the method described above. The infection of each mutant was confirmed by RT-PCR using primers EAPV-CP fwd and rev (Table 1).

### 2.5. Time Course of Virus Accumulation by Enzyme-Linked Immunosorbent Assay (ELISA)

To monitor the accumulation levels of mutated viruses of EAPV-d6I_8_ and EAPV-d6 in plants, indirect ELISA using the antiserum against EAPV CP [19] was performed according to the method previously described [41]. Nine diameter leaf discs, 0.5 cm each, were punched from three different upper leaves of each infected passionfruit plant at 5, 10, 15, 20, 25, and 30 dpi. The discs from each plant were combined as one sample, and triplicate samples were detected by indirect ELISA with EAPV CP antiserum [19]. Reactions were measured as absorbance at 405 nm using a Multiskan FC Microplate Reader (ThermoFisher, Singapore).

### 2.6. Genetic Stability of Mild Strain EAPV-d6I_8_

The stability of attenuated mutant EAPV-d6I_8_ was assessed in passionfruit plants through mechanical inoculation with EAPV-d6I_8_. A 10 μL aliquot of the mutant plasmid was applied to each *C. quinoa* leaf, which had been dusted with Carborundum, using cotton swabs. Subsequently, viruses from the infected *C. quinoa* leaves were transferred to yellow passionfruit plants seven days post-inoculation (dpi). Serial passaging of the virus from infected passionfruit plants to fresh, uninfected plants was conducted every three weeks. After six successive passages, the stability of EAPV-d6I_8_ in 20 plants was verified by sequencing the HC-Pro fragment amplified by RT-PCR, as described above.

### 2.7. Cross-Protection Assay of the Attenuated Mutant EAPV-d6I_8_ against the Taiwan Severe Strain EAPV-TW-GFP

The cross-protection effectiveness of the attenuated mutant EAPV-d6I_8_ was conducted on the plants of *N. benthamiana* and yellow passionfruit against the challenge of a recombinant of Taiwan isolate EAPV-TW-GFP, investigated under temperature-controlled greenhouse (25 to 28 °C) conditions. One-month-old *N. benthamiana* plants were mechanically inoculated with the attenuated mutant EAPV-d6I_8_. Ten days after the protective inoculation, the infection of *N. benthamiana* plants was verified by RT-PCR with EAPV CP primer pair (Table 1). Then, two upper fully expanded leaves were mechanically challenged with EAPV-TW-GFP, prepared from 0.5 g of infected *N. benthamiana* leaves using 2 mL of 0.01 M potassium phosphate buffer (pH 7.2). Protection was evaluated by symptom development in the challenged plants at 21 days post-challenge (dpc) and fluorescent signals under UV light. The presence or absence of the challenge virus was also confirmed by RT-PCR with the primer pair EAPV 2/EAPV HCPro rev (Table 1) that was directed toward genomic regions flanking the GFP reading frame.

The same approach was applied to yellow passionfruit plants. Virus-free yellow passionfruit seedlings were prepared through cutting. EAPV-d6I_8_, derived from 0.5 g of infected *N*. *benthamiana* leaves, were mechanically introduced to the upper two fully expanded leaves of yellow passionfruit plants two weeks after rooting. Ten days following the protective inoculation, the two uppermost fully expanded leaves were individually challenged with EAPV-TW-GFP, each prepared from 0.5 g of leaf tissue from infected *N. benthamiana* plants, as previously described. Protection efficacy was assessed by observing symptom development in the systemic leaves and detecting the challenge virus through RT-PCR at 21 days post-challenge, following the procedures described above.

### 2.8. Cross-Protection Evaluation of the Attenuated Mutant EAPV-d6I_8_ against the Vietnam Severe Strain EAPV-GL1

A similar assay was also performed in yellow passionfruit plants protected by EAPV-d6I_8_ against the Vietnam severe strain EAPV-GL1. Protection was evaluated by the development of symptoms in challenged plants at 21 dpc. The presence or absence of the challenge virus was verified by RT-PCR with primers GL1 9042 fwd (a specific primer for the GL1 isolate) and EAPV CP rev (Table 1). Each cross-protection assay was conducted with ten plants, and the experiment was repeated three times.

## 3. Result

### 3.1. Infectivity of EAPV-TW-GFP

The EAPV-TW was labeled with GFP between the P1 and HC-Pro cistrons (Figure 1A) to monitor the severe strain in challenged plants during cross-protection tests. The recombinant virus EAPV-TW-GFP, in vivo derived from the infectious clone pEAPV-TW-GFP, induced leaf curling and leaf mosaic symptoms in *N. benthamiana* plants at 14 dpi. Additionally, GFP was observed in the systemic upper leaves of these plants (Figure 1B). Yellow passionfruit plants inoculated with progeny viruses obtained from *N. benthamiana* infected with pEAPV-TW-GFP displayed leaf curling, leaf mosaic, and fluorescent signals on systemic upper leaves at 14 dpi (Figure 1B). The symptoms caused by pEAPV-TW-GFP in both hosts were similar to those induced by EAPV-TW (Figure 1B). EAPV-TW-GFP and EAPV-TW have equivalent infectivity (100%) through mechanical inoculation on *N. benthamiana* and yellow passionfruit plants.

GFP and viral sequences were detected in the infected passionfruit plants using RT-PCR with primers EAPV P1 fwd and EAPV HC-Pro rev. The results showed that the 1036 bp product was amplified in the plants infected with EAPV-TW-GFP, while a shorter product of 280 bp was amplified in the plants infected with native EAPV-TW (Figure 2A). The expression of GFP in passionfruit plants was further verified by Western blotting using GFP antiserum [40] and EAPV CP antiserum [19] (Figure 2B).

### 3.2. Symptoms Induced by Mutants EAPV-d6 and EAPV-d6I_8_ in N. Benthamiana and Yellow Passionfruit Plants

The N-terminal region of EAPV HC-Pro was modified with a deletion of six aa upstream of the FFKG to generate EAPV-d6 or further coupled with F_8_I to generate EAPV-d6I8 (Figure 3A). The mutant EAPV-d6 or EAPV- d6I_8_ was mechanically introduced into *N. benthamiana* and yellow passionfruit plants. Symptoms induced by individual EAPV mutants in plants of *N. benthamiana* and yellow passionfruit compared to those caused by the wild-type EAPV-TW are shown in Figure 3B. In *N. benthamiana* leaves, EAPV-d6 showed mild mottling symptoms, and in yellow passionfruit leaves, it induced slight distortion symptoms and mild mosaic. In contrast, EAPV-d6I_8_ did not show visible symptoms on the leaves of both *N. benthamiana* and yellow passionfruit plants (Figure 3B). For confirmation of mutated viruses, the entire HC-Pro of individual mutants was amplified by RT-PCR with primers EAPV 2 fwd and EAPV P3 rev (Table 1). Sequencing the PCR products confirmed the presence of mutations in the mutated viruses. Infection of EAPV-d6 or EAPV-d6I_8_ in passionfruit plants was also confirmed by RT-PCR with EAPV CP primers (Table 1, Figure 3C).

**Figure 3 viruses-16-01231-f003:**
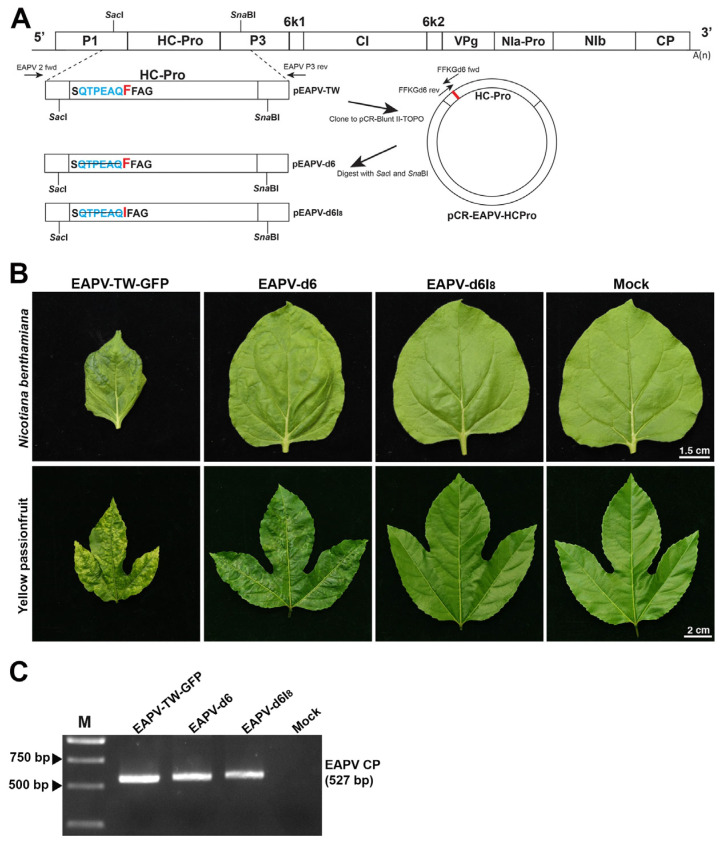
Symptoms of individual mutants derived from East Asian Passiflora virus Taiwan strain TW (EAPV-TW) in plants of *Nicotiana benthamiana* and yellow passionfruit. (**A**) Schematic representation of two EAPV-TW mutants constructed from the infectious clone of pEAPV-TW. Deletion mutations of the N-terminal six amino acids alone (d6) or coupled with F_8_I mutation (d6I_8_) are indicated. (**B**) Different symptoms were induced in *N. benthamiana* and yellow passionfruit leaves at 14 days after inoculation with EAPV-d6 or EAPV-d6I_8_, as compared with those induced by the wild-type EAPV-TW-GFP. (**C**) Infectivity of individual mutants was confirmed by RT-PCR with EAPV CP forward and reverse primers. Lane M, 10,000-base pairs DNA ladder.

### 3.3. Accumulation Dynamics of EAPV Mutants

The indirect ELISA with EAPV CP antiserum was used to monitor the time–course accumulation levels of EAPV-TW-GFP and individual mutants in inoculated *N. benthamiana* plants at 5, 10, 15, 20, 25, and 30 dpi. The accumulation levels of both EAPV-d6 and EAPV-d6I_8_ increased from 5 to 10 dpi and decreased sharply from 10 to 15 dpi (Figure 4). CP levels of mutant EAPV-d6I_8_ were significantly lower than EAPV-d6. After 15 days, CP levels of EAPV-d6 were maintained at relatively higher levels than EAPV-d6I_8_, which displayed a low-level up-and-down equilibrium (Figure 4).

### 3.4. The Stability of Attenuated Strain in Passionfruit Plants

EAPV-d6I_8_ was tested for the stability test in yellow passionfruit plants through successive passages under temperature-controlled greenhouse conditions. The EAPV mutant still induced attenuated symptoms after six successive mechanical inoculations, each with a three-week interval. According to the sequencing of the individual cDNA fragments of HC-Pro amplified from infected passionfruit plants as described above, the deletion and the point mutation of d6I_8_ remained the same.

### 3.5. Cross-Protection Effectiveness of EAPV Mutants against Taiwan EAPV Severe Strain

EAPV-d6I_8_, which did not induce conspicuous symptoms in *N. benthamiana* and yellow passionfruit plants, was selected as a candidate for the cross-protection tests. The Taiwan severe virus EAPV-TW-GFP was used to challenge the *N. benthamiana* and yellow passionfruit plants protected by EAPV-d6I_8_. The unprotected (mock) plants showed severe leaf mosaic and distortion in both *N. benthamiana* and yellow passionfruit plants at 21 dpc (Figure 5A). In contrast, all the *N. benthamiana* and yellow passionfruit plants protected by EAPV-d6I_8_ and challenged with EAPV TW-GFP did not show visual symptoms at 21 dpc (Figure 5A), and these effects of protection were observed for up to two months in both plants. Furthermore, no GFP fluorescence was detected in all the protected plants, while mock plants displayed strong fluorescent signals when challenged with EAPV-TW-GFP at 21 dpc. The amplified fragments of 1036 bp and 280 bp were from unprotected and protected plants, respectively, after being challenged with EAPV-TW-GFP (Figure 5C).

Our results indicated that the mutant EAPV-d6I_8_ confers complete protection (100%) against Taiwan severe strain EAPV-TW-GFP in both *N. benthamiana* and yellow passionfruit plants.

**Figure 5 viruses-16-01231-f005:**
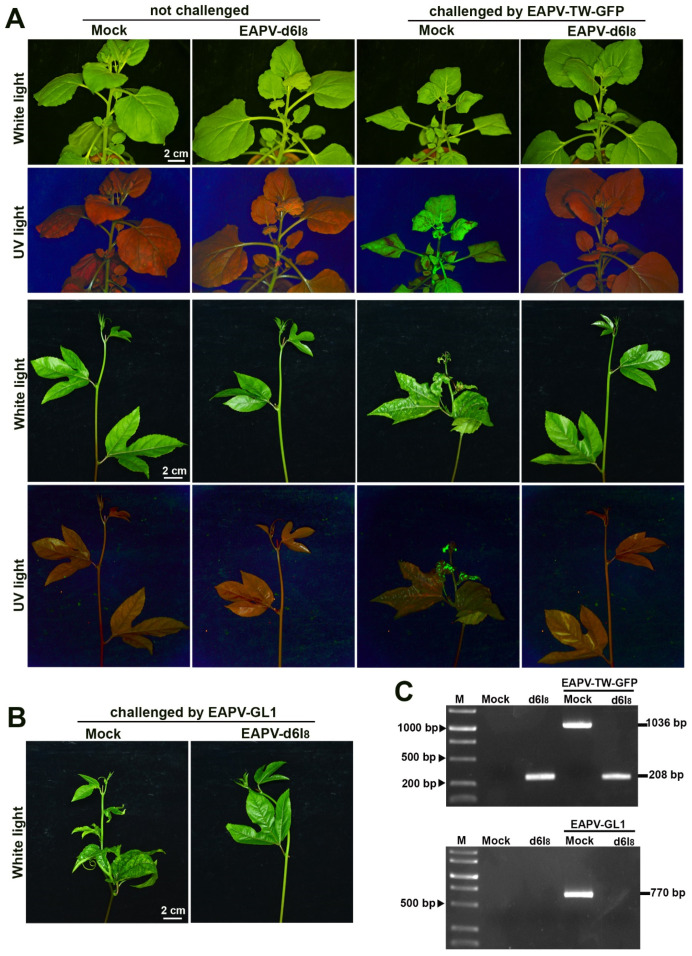
Cross-protection effectiveness of East Asian Passiflora virus (EAPV)-d6I_8_ against the challenge inoculation with the severe strain EAPV-TW-GFP (**A**) in *Nicotiana benthamiana* and yellow passionfruit plants and EAPV-GL1 (**B**) in yellow passionfruit plants. All plants were challenged with EAPV-GFP or EAPV-GL1 ten days after the protective inoculation with EAPV-d6I_8_. Symptoms in *N. benthamiana* and passionfruit plants were photographed 14 days or 21 days, respectively, after the challenge inoculation under white light and UV light. (**C**) RT-PCR detection of EAPV-TW-GFP or EAPV-GL1 in upper no inoculated passionfruit leaves at 21 days post-challenge inoculation. The fragments of 1036 bp and 280 bp were amplified by EAPV 2 fwd/EAPV HCPro rev to detect EAPV-TW-GFP. To specifically detect EAPV-GL1, the fragment of 770 bp was amplified by GL1 9042 fwd/EAPV CP rev. Lane M, 10,000-base pairs DNA ladder.

### 3.6. Cross-Protection Effectiveness of EAPV Mutants against Vietnam EAPV Severe Strain

In addition to the protection against Taiwan EAPV-TW-GFP, the attenuated mutant EAPV-d6I_8_ was further used to protect *N. benthamiana* and yellow passionfruit plants against the Vietnam severe strain EAPV-GL1. After 21 days of the challenge inoculation, all unprotected plants displayed severe symptoms, and protected plants showed no apparent symptoms (Figure 5B). Using the primers GL1 9042 fwd (Table 1, a specific primer for the GL1 isolate) and EAPV CP rev (Table 1), EAPV-GL1 was only detected in unprotected plants as an amplified product of 770 bp but not in the protected plants at 21 dpc (Figure 5C). Our results indicated that the mutant EAPV-d6I_8_ also confers complete protection (100%) against Vietnam severe strain EAPV-GL1 in *N. benthamiana* and yellow passionfruit plants.

## 4. Discussion

The infectious clone of pEAPV-TW was tagged with GFP between P1 and HC-Pro cistrons for monitoring the virus in challenged plants during cross-protection assay. *N. benthamiana* and yellow passionfruit plants infected with pEAPV-TW-GFP showed similar symptoms to those infected with the native virus. The expression of GFP in plants infected with EAPV-TW-GFP was demonstrated with fluorescent signals under UV light (Figure 1B) and Western blotting analysis (Figure 2B). The insertion of GFP to pEAPV-TW was similar to our previous studies for potyviruses of ZYMV (Hsu et al. 2004), TuMV [42], and PaMoV [43].

In this study, the deletion of six amino acids upstream of the F_8_FKG motif in the N-terminal region of HC-Pro was performed individually or simultaneously with a point mutation F_8_I to generate two mutants EAPV-d6 and EAPV-d6I_8_, respectively. The mutant EAPV-d6 and a previous generation mutant EAPV-I_8_ [39] still exhibited apparent symptoms in passionfruit plants, with only a slight reduction in viral symptoms. However, the combination of a small deletion d6 coupled with F_8_I generated EAPV-d6I_8_ that did not induce visible symptoms in passionfruit plants (Figure 3B). These results were similar to previous studies in the highly conserved F_8_WKG motif (FFKG in EAPV) in FWKG-α-helix 1 element of TuMV, ZYMV [35], and PRSV [38]. FWKG-α-helix 1 element was reported to affect the RSS capability and self-interaction of HC-Pro [35]. The combination of deletion and point mutation in TuMV and PRSV was demonstrated to be more effective in reducing potyviral pathogenicity than single point mutation or upstream deletion [35,38].

The attenuated mutant EAPV-d6I_8_ was selected for the cross-protection test because its infection in *N. benthamiana* and yellow passionfruit plants resulted in no apparent symptoms (Figure 3B). In ELISA assays, the time–course CP accumulation levels of EAPV-d6I_8_ showed a zigzag pattern. The attenuated virus accumulated in the hosts at high levels until ten dpi and then gradually declined thereafter to display a zigzag pattern of accumulation, indicating it can trigger the host defensive RNA silencing, and the reduced capability for suppressing the host RNA silencing resulted in the sharp drop and followed by an up-and-down low-level equilibrium. This phenomenon has been shown in previous studies as an important criterion for beneficial protective viruses of ZYMV [28], TuMV [34,35], and EAPV [39]. In the case of EAPV-d6, its accumulation levels also displayed a sharp drop at ten dpi, but it was maintained in relatively higher accumulation levels from ten to 30 dpi (Figure 4), which may explain their symptomatic phenotype in *N. benthamiana* and yellow passionfruit plants.

The FWKG-α-helix 1 element was reported to affect the functions of HC-Pro self-interaction and aphid transmission of TuMV [35]. The lack of aphid transmission ability of the TuMV-attenuated mutant containing a point mutation and a small deletion in the FWKG-α-helix 1 element is due to the decrease in HC-Pro self-interaction [35]. Similar to TuMV, EAPV-d6I_8_ is expected as a non-aphid-transmissible attenuated mutant, an important criterion for a useful protective virus [44]. Nevertheless, its vector transmissibility remains to be further verified.

Plant virus stability is one of the essential criteria for the successful application of mild strains [44]. Sometimes, a single-point mutation can reverse a mild virus to a virulent phenotype [45]. Thus, a mild strain generated by single- or double-point mutation still poses a risk of reverting back to a severe strain when applied in the field. In our previous study, two effective mild EAPV mutants were generated by double-point mutation [39]. Here, EAPV-d6I_8_ contains not only a single-point mutation but also an 18-nucleotide deletion (6 aa), which is predicted to be more difficult to reverse back to the wild type and should be more genetically stable than single- or double-point mutants.

Another criterion for an effective mild strain is that it should not negatively interact with other viruses in hosts [46]. The tip necrosis disease in passionfruit protected by a mild strain of PWV in Australia [47] indicated a breakdown in protection. This severe symptom was demonstrated as a synergistic effect between PWV and cucumber mosaic virus (CMV). CMV was found to infect passionfruit in Taiwan [48]. Moreover, the dual infection of EAPV and CMV showed more severe symptoms than individual infection in yellow passionfruit plants under greenhouse conditions. Hence, a bivalent protective virus, EAPV-d6I_8_ harboring CMV CP, is being developed to reduce this potential risk. In addition, the coinfection of EAPV and another potyvirus-like PaMoV in passionfruit plants was recorded [3]. Recently, we have shown that the combination of a two-in-one attenuated vaccine of papaya ringspot virus (PRSV) and zucchini yellow mosaic virus (ZYMV) confers concurrent protection against the two unrelated potyviruses in cucurbits [49]. The combination of the attenuated mutant EAPV-d6I_8_ developed in this study and our previously developed PaMoV-EI [43], which is an attenuated strain of PaMoV, promises to protect passionfruit plants against the two major potyviruses of EAPV and PaMoV in passionfruit plants simultaneously.

## Figures and Tables

**Figure 1 viruses-16-01231-f001:**
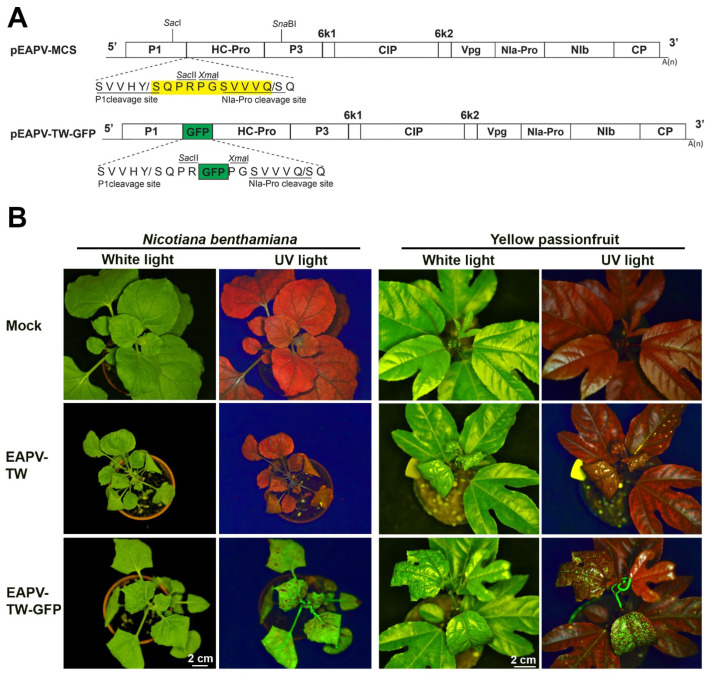
Infectivity of East Asian Passiflora virus (EAPV) infectious clone tagged with green fluorescent protein (GFP) in *Nicotiana benthamiana* and yellow passionfruit plants (*Passiflora edulis* f. *flavicarpa*). (**A**) Schematic representation of the portion of the genomic region of EAPV-TW to insert the *GFP* reading frame. The yellow box indicates the insertion cassette containing multiple cloning sites (MCS: *Sac*II, *Xma*II) and an NIa-Pro cleavage site (SVVVQ/S). The GFP reading frame (green box) was inserted between P1 and HC-Pro genes utilizing the *Sac*II and *Xma*I sites. (**B**) Detection of fluorescent signals emitted by EAPV-TW-GFP under UV light in leaves of *N. benthamiana* and yellow passionfruit.

**Figure 2 viruses-16-01231-f002:**
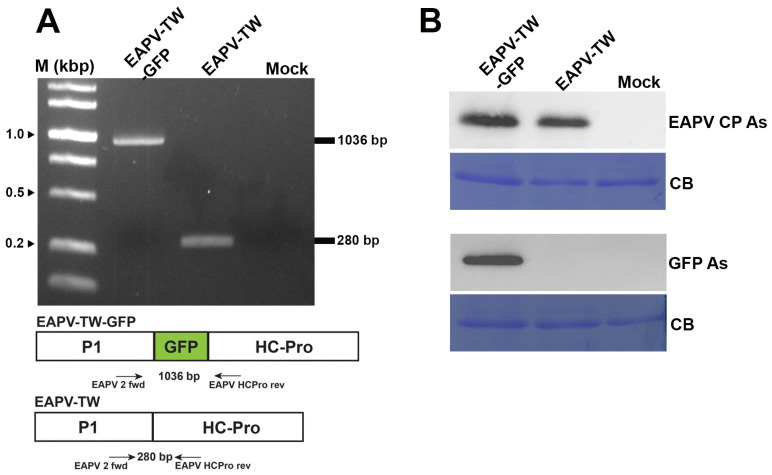
Detection of GFP gene or protein in *Nicotiana benthamiana* inoculated with pEAPV-GFP. (**A**) Infectivity of pEAPV-GFP was confirmed by RT-PCR. The primer pair position indicated by arrows was used for RT-PCR. Primer pair EAPV 2 fwd/EAPV HCPro rev amplified 1036-bp product of EAPV-TW-GFP and 280-bp product of EAPV-TW. Lane M, 10,000-base pairs DNA ladder. (**B**) Western blotting analysis with EAPV CP antiserum (EAPV CP As) and GFP antiserum (GFP As). Mock plants were inoculated with phosphate buffer only, and Coomassie blue-stained (CB) total protein was used as the loading control.

**Figure 4 viruses-16-01231-f004:**
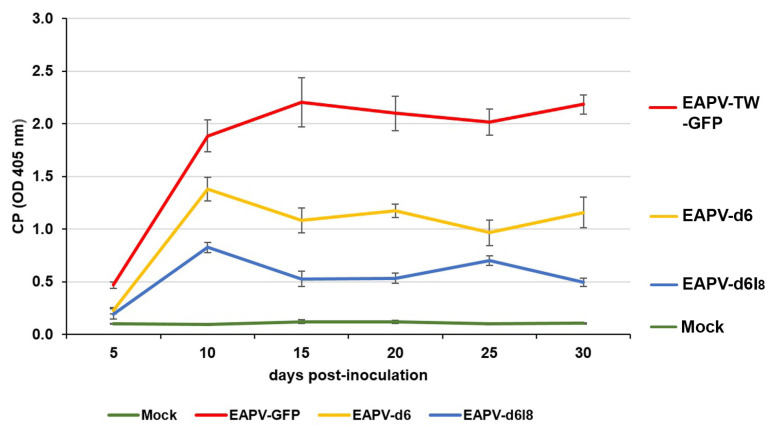
Time–course accumulation levels of coat protein (CP) of the recombinant EAPV-TW-GFP or individual East Asian Passiflora virus (EAPV) mutants in inoculated *Nicotiana benthamiana* plants as determined by ELISA with EAPV CP antiserum [19]. Each absorbance value represents the average reading from three replicates. Plants infected with the wild-type EAPV-TW and inoculated with buffer were used as positive and negative controls, respectively.

**Table 1 viruses-16-01231-t001:** Oligonucleotide primers used in this study.

Name	Sequence 5′-3′	Underline Note
Cloning primer		
EAPV 2 fwd	GAGGACGTATGGGAACAAAGTTAG	
P1HCPro fwd	GCCCCCGGG**GACTCAGTTGTTGTCCAG**TCACAAACACCTGAGGCTCA	*Xma*I
P1HCPro rev	AACTGAGTCCCCGGGGGCCCGCGGTGTTTGTGAGTAGTGCACAACAC	*Xma*I, *Sac*II
EAPV P3 rev	CAAGCTTAGCGCGTGCCACTCCTG	
GFP SacII fwd	ACACCGCGGATGGTGAGTAAAGGAGAAGAACTT	*Sac*II
GFP XmaI rev	AGTCCCGGGTTTGTATAGTTCATCCATGCC	*Xma*I
EAPV P1 fwd	GATGAACAAATCAAGCGTGGTGA	
EAPV HCPro rev	CCCCACATTGAGCATTCGCAAAA	
Site direct mutagenesis primers		
EAPVd6 fwd	GTGCACTACTCATTTTTCGCCGGCTGGAAGAAG	
EAPVd6 rev	GGCGAAAAATGAGTAGTGCACAACACTCTCTTT	
EAPVd6F8I fwd	ATTTTCGCAGGATGGAAGAAGGTTTTTGATAGG	
EAPVd6F8I rev	CTTCTTCCATCCTGCGAAAATTGAGTAGTGTAC	
Detection		
EAPV CP fwd	CCCATGGTATTCAGCAGTCCAAAGAG	
EAPV CP rev	GGGAATTCATACCCAAAAGGGTGT	
GL1 9042 fwd	CACAACCCCGATAAATCCAAATC	

Restriction enzyme sites are underlined with black lines. Bold characters indicate the NIa cleavage site. All primers are designed based on sequence of EAPV-TW (KP114136) except primer GL1 9042 fwd, which is designed based on sequence of EAPV-GL1 (MT450870).

## Data Availability

All data have been made available in the manuscript.

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
