# Peer review of "The Generation of Attenuated Mutants of East Asian Passiflora Virus via Deletion and Mutation in the N-Terminal Region of the HC-Pro Gene for Control through Cross-Protection"

_viruses, 2024, doi:10.3390/v16081231_

Round 1
Reviewer 1 Report
Comments and Suggestions for Authors
Dear authors,
The attenuated mutant that your group has generated has potential use for controlling passionfruit woodiness disease caused by the East asian passiflora virus and represents an important technological contribution. The manuscript has a logical and coherent structure, with most methodologies well described. However, I consider some areas that need to be addressed to improve precision and clarity.
Major issues include: 1) Removal and restructuring of an entire paragraph at the end of the introduction that replicates the abstract. 2) Relocation of three figures to appropriate sections. 3) Confirmation of the names and sequences of some primers that appear non-specific to the virus used in the study. 4) Improvement of Table 1. 5) Inclusion of the number of plants evaluated in the cross-protection assay to enhance the precision and significance of the results in proportion to the sample size.
These and other minor suggestions and comments that need to be addressed have been indicated in the manuscript.
I hope that will be useful to improve the quality of your manuscript.
Best regards

Reviewer 2 Report
Comments and Suggestions for Authors
1. In Fig 1B, what’s the Latin name of yellow passionfruit? How many days after inoculation of the inoculation that the images displayed? Is there have the same infectivity rate of the EAPV-TW and EAPV-TW-GFP on the N. benthamiana and yellow passionfruit?
2. Fig 2 is the detection results of the Fig 1, please integrated these two figures together.
3. In Fig 2A and 2B, the molecular size of the DNA and proteins are marked, please add it in the Figures.
4. Please add statistical data of the infectivity ratio of EAPV-TW and EAPV-TW-GFP on the N. benthamiana and yellow passionfruit in Fig 2.
5. Please adding the description of why the author chose to deletion of the six aa and mutation of the I8? How did the author know that mutation of these sites on HCpro could attenuate the viral pathogenicity on these two plants?
6. In Fig 3C, please add the DNA marker on the gel. Why the PCR products of the EAPV-d6 and EAPV-d6I8 have a larger size than that of the EAPV-TW-GFP? According to your description, these two mutants have a 6 amino acid deletion, and its coding sequence are 12 nt shorter than the EAPV-TW-GFP when using the same pairs of primer in PCR. Please explain.
7. Please add the immune-blot detection of the virus using anti-CP antibodies for demonstrating that the systemic infection of the EAPV-d6 and EAPV-d6I8 on N. benthamiana and yellow passionfruit. RT-PCR detection often have false results that facing the fact that Agrobacterium-expressed RNAs could trafficking in plant and cause contaminations, not systemic movement of the EAPV.
8. In Fig 4, please revise the chart. Add the abscissa axes and ordinate axes, add the statistical analyses and significance analyses.
9. In Fig 5C, please add the immune-blot detection of the virus using anti-CP antibodies. Please add a chart for description of the cross-protection rate of the EAPV-d6I8 mutant to EAPV-TW, EAPV-TW-GFP, and EAPV-GL.
10. How does the RNA silencing suppression activity change of the HCPro-d6I8 and HCPro-d6? Please add these results in the MS.
11. The author published several pieces of the MS descripting of the cross-protection using the attenuated virus by directly mutation viral VSR. I am wondering how many mutants obtained that have stable and effective cross-protection roles in the field? As we know that all artificial mutations on the VSR are not stable, it’s easy to occur reverse mutation and back to wild-type. Hence, please statistic the reverse mutation rate of your EAPV-d6 and EAPV-d6I8 on the systemic infected leaves of N. benthamiana and yellow passionfruit.
Round 2
Reviewer 2 Report
Comments and Suggestions for Authors
No